# Clinical characteristics of sarcoma cases in which long-term disease control was achieved with trabectedin treatment: A retrospective study

**Akihiro Ohmoto**[1], **Kenji Nakano**[1], **Naoki Fukuda**[1], **Xiaofei Wang**[1], **Tetsuya Urasaki**[1], **Naomi Hayashi**[1], **Hirotaka Suto**[1], **Shohei Udagawa**[1], **Ryosuke Oki**[1], **Yasuyoshi Sato**[1], **Mayu Yunokawa**[1], **Makiko Ono**[1], **Masanori Saito**[2], **Yusuke Minami**[2], **Keiko Hayakawa**[2], **Taisuke Tanizawa**[2], **Keisuke Ae**[2], **Seiichi Matsumoto**[2], **Junichi Tomomatsu**[1], **Shunji Takahashi**[1] *

1 Department of Medical Oncology, The Cancer Institute Hospital of the Japanese Foundation for Cancer Research, Tokyo, Japan, 2 Department of Orthopedic Oncology, The Cancer Institute Hospital of the Japanese Foundation for Cancer Research, Tokyo, Japan

* s.takahashi-chemotherapy@jfcr.or.jp

**Data Availability Statement:** All relevant data are within the paper and its Supporting information files.

## Abstract

Trabectedin is a therapeutic option for patients with advanced sarcoma. While a randomized trial demonstrated its prolonged progression-free survival (PFS), the reported PFS was <6 months. Some patients can achieve long-term disease control with this treatment. However, the reference information is insufficient. Herein, we retrospectively reviewed 51 sarcoma patients who received trabectedin. We analyzed the clinicopathological features, trabectedin dose, administration schedule, and clinical outcomes, including the overall response rate (ORR) and PFS. Among them, we assessed the detailed data of patients who achieved long-term disease control (PFS >1 year). The ORR in the 49 evaluable patients was 8%, and the median PFS in 51 patients was 7.5 months. Six patients (12%) achieved PFS of >1 year. Five of the six patients had metastatic lesions at trabectedin initiation. The pathological subtypes were myxoid liposarcoma (n = 2), leiomyosarcoma (n = 2), synovial sarcoma (n = 1), and Ewing sarcoma (n = 1). The final administration dose was the minimum dose (0.8 mg/m$^2$) in two patients who continued the treatment over 20 cycles. The best radiological response was partial response (PR) in two myxoid liposarcoma patients and stable disease in four. The durations from trabectedin initiation to the first response in the two PR cases were 163 and 176 days, respectively. Our results support the validity of continuing trabectedin at a sustainable dose and interval in patients who can tolerate it. These results may be useful when considering the clinical application of trabectedin.

## Introduction

Limited treatment options are available for patients with advanced or metastatic sarcoma after anthracycline failure. According to the National Comprehensive Cancer Network guidelines

**Funding:** The authors received no specific funding for this work.

**Competing interests:** NO authors have competing interests.

for advanced/metastatic soft tissue sarcoma, trabectedin is considered a second- or later-line chemotherapy drug along with pazopanib and eribulin [1]. This agent is classified into category 1 for liposarcoma and leiomyosarcoma and category 2A for other subtypes. Trabectedin is an alkylating agent that binds to DNA and blocks DNA repair mechanisms [2]. This agent has been assessed, especially for translocation-related sarcoma because trabectedin mechanistically suppresses the transcription of the oncogenic fusion proteins, such as *FUS-DDIT* [3]. A French randomized phase II trial demonstrated the progression-free survival (PFS) benefit of trabectedin continuation until disease progression after the first six cycles [4]. Another randomized phase II trial comparing trabectedin and best supportive care demonstrated significant PFS prolongation with trabectedin (hazard ratio 0.07, $P < 0.0001$) [5]. However, the median PFS was <6 months, and the overall response rate (ORR) was only 8%. Therefore, further investigations are warranted to maximize clinical benefits. Patients with myxoid liposarcoma harboring the *FUS-DDIT* gene fusion are considered promising candidates for trabectedin. A retrospective study of 51 patients with pretreated myxoid liposarcoma exhibited remarkable ORR (51%) and PFS (median 14.0 months) [6].

Despite its importance, few clinical predictive factors of trabectedin's benefits are known.

We previously published our clinical experience of 38 patients with advanced sarcoma treated with trabectedin [7]. Our current study includes a larger sample size, and it highlights the clinical features of patients with long-term disease control achieved with trabectedin.

## Materials and methods

This study was approved by the Institutional Review Board of the Japanese Foundation for Cancer Research (2021-GB-088), and was conducted in accordance with the precepts established by the Helsinki Declaration. The requirement for informed consent was waived by the institutional review board. We retrospectively reviewed our institutional clinical database from 2012 to 2021 and extracted the data of 51 advanced sarcoma patients who received trabectedin. Clinical data included sex, age at trabectedin initiation, Eastern Cooperative Oncology Group performance status (PS), clinical stage, primary site, metastatic site (for metastatic cases), pathological features (histological subtype, Fédération Nationale des Centres de Lutte Contre le Cancer [FNCLCC] grade) [8], and presence of translocation. Data regarding chemotherapy included any anti-cancer agent used before or following trabectedin, the trabectedin dose, and the administration schedule. Trabectedin was approved in Japan in 2015, and our analysis included patients who received it as a regular prescription and as part of a clinical trial prior to regulatory approval. Available data were also collected for patients who had undergone next-generation sequencing (NGS) analysis or microsatellite instability (MSI) testing using resected/biopsied tumor tissues. The GEISTRA score proposed by the Spanish sarcoma research group is an index associated with prognosis in trabectedin-treated patients [9]. This score was calculated using three indices (non-L-sarcoma, metastatic-free interval from initial diagnosis <8.1 months, and Karnofsky PS <80).

Clinical outcomes included radiological ORR, duration from trabectedin initiation to the first radiological response, PFS, and overall survival (OS). PFS and OS were calculated from trabectedin initiation, and PFS and OS curves were estimated using the Kaplan–Meier method. Univariate analyses of significant factors for PFS and OS were performed using the Cox proportional-hazards regression model, and these were considered statistically significant if two-sided p-values were <0.05. All statistical analyses were performed using EZR version 1.53 (Saitama Medical Center, Jichi Medical University), which is based on R and R commander (http://www.jichi.ac.jp/saitama-sct/SaitamaHP.files/download.html) [10].

## Results

### Patient characteristics and treatment information

The characteristics of 51 patients with sarcoma who had received trabectedin are summarized in Table 1. At trabectedin initiation, the median patient age was 50 (range, 21–74) years, and the PS was 0, 1, and 2 in 33, 17, and one patients, respectively. Forty-seven (92%) patients had a history of treatment with doxorubicin. Common histological subtypes included myxoid liposarcoma (n = 13), synovial sarcoma (n = 10), leiomyosarcoma (n = 6), and dedifferentiated liposarcoma (n = 5). The percentage of L-sarcomas (liposarcoma or leiomyosarcoma) was 47%. The FNCLCC grades were grade 1 (n = 5), grade 2 (n = 18), and grade 3 (n = 14). GEISTRA scores were 0 (n = 18), 1 (n = 23), and 2 (n = 10). Twenty-six (51%) patients harbored any gene fusion, including *FUS-DDIT3* diagnostic for myxoid liposarcoma (n = 10) and *SS18–SSX* diagnostic for synovial sarcoma (n = 8). The MSI status was assessed in five patients, all of whom were MSI-negative. Commercially available NGS (FoundationOne®CDx) was conducted in three cases, and the tumor mutational burden in one case was high (11 mut/Mb). Unfortunately, no druggable mutations were detected in these patients.

Fourteen patients received trabectedin as second-line, 18 as third-line, and 19 as a later-line regimen. The median duration from the initial diagnosis to trabectedin initiation was 37.3 months. Fifty patients received an initial dose of 1.2 mg/m$^2$, and dose reduction during their course was performed in 19 patients (final administration dose: 1.0 mg/m$^2$ in 16 patients and 0.8 mg/m$^2$ in three patients). Due to myelosuppression, one patient received an initial dose of 1.0 mg/m$^2$. The median number of administration cycles until trabectedin termination was 6 (range, 1–26), and two patients continued the treatment over 20 cycles. Twenty-eight (55%) patients received any chemotherapy, including pazopanib (n = 24) and eribulin (n = 8), after trabectedin termination.

### Clinical outcomes

The ORR in 49 evaluable patients was 8%, and that in 13 myxoid liposarcoma cases was 23%. For responders, the median duration from trabectedin initiation to radiological response was 170 (range, 33–198) days. The median PFS in all patients was 7.5 months, and the 6-month and 1-year PFS rates were 57% and 23%, respectively (Fig 1A). Subgroup analysis showed that patients with myxoid liposarcoma tended to have longer PFS than those with other histological subtypes (median PFS, 11.7 months vs. 6.5 months, *P* = 0.054) (hazard ratio [HR]: 0.42, 95% confidence interval [CI]: 0.17–1.04, *P* = 0.06) (Fig 1B). Between four patients with partial response (PR) and 35 patients with stable disease (SD), PFS was favorable in the former (median PFS, 11.7 months vs. 9.1 months, *P* = 0.09) (HR: 0.29, 95% CI: 0.06–1.30, *P* = 0.10) (Fig 1C). However, cases with FNCLCC grades 1, 2, and 3 did not significantly differ (*P* = 0.37), nor did cases with GEISTRA scores 0, 1, and 2 (*P* = 0.61), translocation-positive and negative cases (*P* = 0.96), or cases that received the treatment as second-line, third-line, or later-lines, respectively (*P* = 0.77). The median OS was 20.2 months, and the 1- and 2-year OS rates were 73% and 39%, respectively (Fig 1D). Unlike PFS, the OS did not significantly differ between myxoid liposarcoma and other histological subtypes (*P* = 0.54).

### Clinical features of patients who achieved long-term disease control (PFS >1 year)

The detailed clinical data of six (12%) patients who maintained disease control over 1 year are shown in Table 2. All six patients had confirmed disease progression prior to trabectedin

**Table 1. Clinical characteristics of 51 sarcoma patients who received trabectedin.**

| Variables | Number (%) |
|---|---|
| Age, years | Median 50 (21–74) |
| Sex | |
| Male | 26 (51%) |
| Female | 25 (49%) |
| ECOG PS | |
| 0 | 33 (65%) |
| 1 | 17 (33%) |
| 2 | 1 (2%) |
| Previous doxorubicin exposure | 47 (92%) |
| Common histological subtype | |
| Myxoid liposarcoma | 13 (25%) |
| Synovial sarcoma | 10 (20%) |
| Leiomyosarcoma | 6 (12%) |
| Dedifferentiated liposarcoma | 5 (10%) |
| Chondrosarcoma | 4 (8%) |
| Undifferentiated pleomorphic sarcoma | 3 (6%) |
| Other histological subtypes* | 10 (20%) |
| FNCLCC grade** | |
| Grade 1 | 5 (14%) |
| Grade 2 | 18 (49%) |
| Grade 3 | 14 (38%) |
| GEISTRA score | |
| Score 0 | 18 (35%) |
| Score 1 | 23 (45%) |
| Score 2 | 10 (20%) |
| Any gene fusion | 26 (51%) |
| Any metastatic lesion at trabectedin initiation | 42 (82%) |
| Treatment line | |
| Second-line | 14 (27%) |
| Third-line | 18 (35%) |
| Later-lines | 19 (37%) |
| Initial trabectedin dose | |
| 1.2 mg/m$^2$ | 50 (98%) |
| 1.0 mg/m$^2$ | 1 (2%) |
| Administration cycle until trabectedin termination | 6 (1–26) |
| Post-trabectedin chemotherapy | 28 (55%) |

ECOG, Eastern Cooperative Oncology Group; PS, performance status; FNCLCC, Fédération Nationale des Centres de Lutte Contre le Cancer.

*Other histological subtypes: alveolar soft part sarcoma (n = 2), endometrial stromal sarcoma (n = 2), rhabdomyosarcoma (n = 1), Ewing sarcoma (n = 1), clear cell sarcoma (n = 1), spindle cell sarcoma (n = 1), undifferentiated endometrial sarcoma (n = 1), and undifferentiated sarcoma (n = 1).

**Japanese grade composed of tumor differentiation, tumor necrosis, and MIB-1 index was available in the other three patients. Grade 2 (n = 1), grade 3 (n = 2).

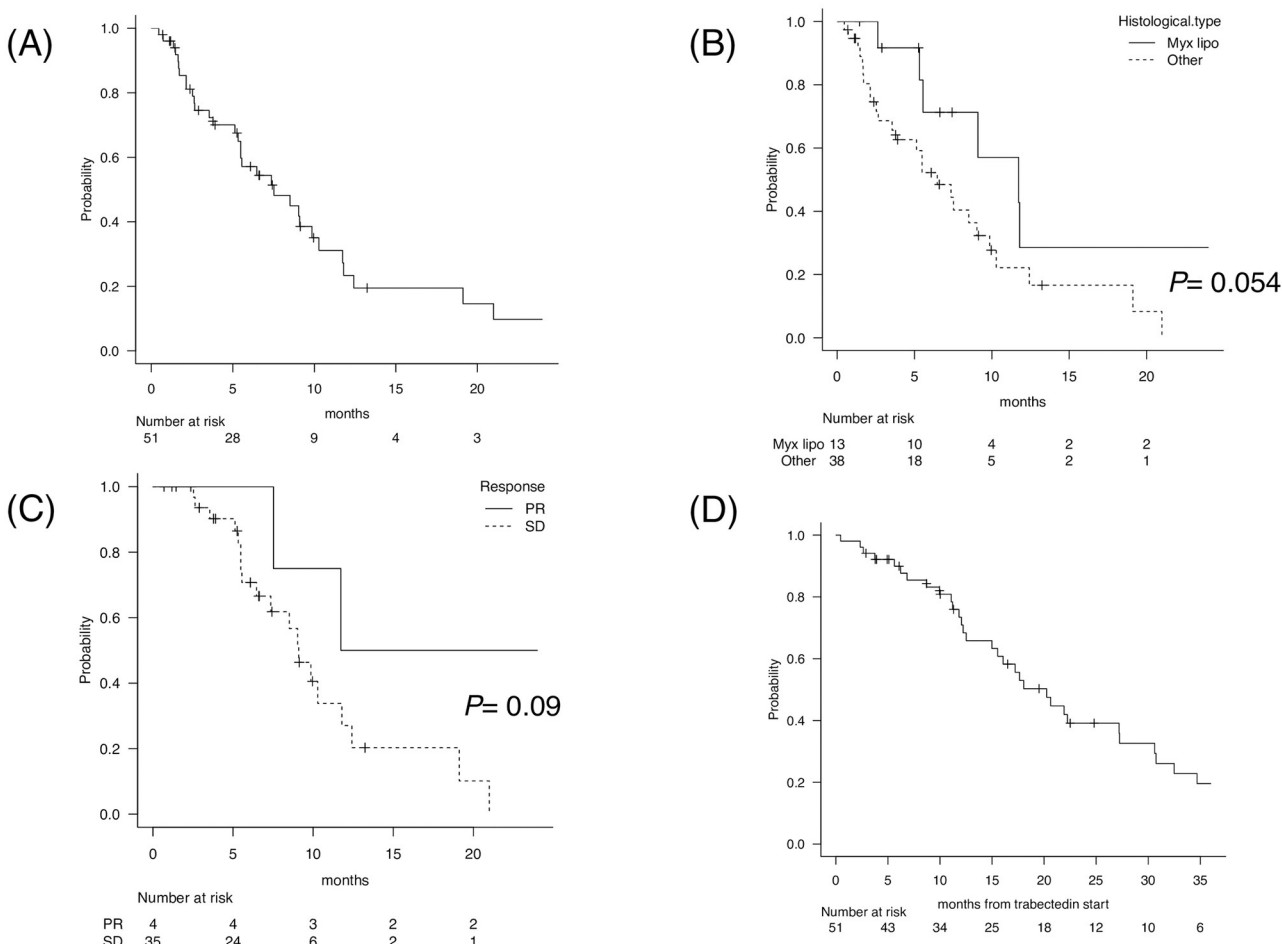

**Fig 1. PFS and OS data in sarcoma patients who received trabectedin.** (**A**) PFS in all 51 sarcoma patients, (**B**) subgroup analysis for PFS between patients with myxoid liposarcoma and those with other histological subtypes, (**C**) subgroup analysis for PFS between patients who achieved PR and SD following trabectedin, and (**D**) OS in all 51 sarcoma patients. PFS, progression-free survival; PR, partial response; SD, stable disease; OS, overall survival; myx lipo, myxoid liposarcoma.

initiation using computed tomography (CT). Five of the six patients had lung metastasis at the start of trabectedin. The pathological subtypes were myxoid liposarcoma (n = 2), leiomyosarcoma (n = 2), synovial sarcoma (n = 1), and Ewing sarcoma (n = 1). The available FNCLCC grades were grades 2 (n = 2) and 3 (n = 1). GEISTRA scores were 0 (n = 3) and 1 (n = 3). Gene fusion characteristics of each subtype were detected in three patients. Trabectedin was adopted as the second-line, third-line, and later-line regimen in one, two, and three patients, respectively. The final administration dose was 1.2 mg/m$^2$ in one patient, 1.0 mg/m$^2$ in three, and 0.8 mg/m$^2$ in two. Interestingly, the final dose in the two patients who continued the treatment over 20 cycles was the minimum dose (0.8 mg/m$^2$). The minimum time interval between trabectedin administration was 3, 4, and 5 weeks in one, four, and one patients, respectively. Two patients with myxoid liposarcoma achieved PR, whereas the best response in the other four patients was SD. After trabectedin initiation, no patients experienced PR following transient progressive disease. The causes of treatment termination included progressive disease (n = 3), liver dysfunction (n = 1), and patient preference (n = 1).

**Table 2. Detailed clinical features in sarcoma patients who achieved long duration (PFS >1 year).**

| Patient ID | Age | Sex | Lesions | Histology | FNCLCC grade | Gene fusion | GEISTRA score | Treatment line | Initial trabectedin dose | Minimum trabectedin dose | Best radiological response | Maximum tumor shrinkage rate from the baseline | Duration from trabectedin initiation to radiological response (for response cases) | Total administration cycle | Cause of termination | PFS from treatment initiation (days) | OS from treatment initiation (days) | OS status |
|---|---|---|---|---|---|---|---|---|---|---|---|---|---|---|---|---|---|---|
| 1 | 47 | M | Lung, vertebra, pancreas, pleura | Myxoid/round cell liposarcoma | NA | NA | 0 | 3 | 1.2 mg/m2 | 0.8 mg/m2 | PR | 68% | 163 days | 24 | Liver dysfunction | 1002 | 1947 | Dead |
| 2 | 31 | M | Lung, vertebra | Ewing sarcoma | NA | EWSR1 | 1 | 2 | 1.2 mg/m2 | 1.0 mg/m2 | SD | Unevaluable (bone lesion) | | 19 | PD | 639 | 1274 | Dead |
| 3 | 70 | F | Lung, liver, soft tissue, iliopsoas | Leiomyosarcoma | grade2 | NA | 1 | 7 | 1.2 mg/m2 | 1.0 mg/m2 | SD | 6% | | 11 | Patient preference | 403 | 936 | Dead |
| 4 | 55 | M | Thigh muscle, groin LN, axilla LN, pancreas | Myxoid/round cell liposarcoma | grade2 | FUS-CHOP type2 | 0 | 5 | 1.2 mg/m2 | 0.8 mg/m2 | PR | 34% | 176 days | 26 | Under treatment | 755 | 755 | Alive |
| 5 | 61 | F | Lung, rib, vertebra, ilium, sacrum | Leiomyosarcoma | NA | NA | 0 | 3 | 1.2 mg/m2 | 1.0 mg/m2 | SD | 22% | | 13 | PD | 378 | 594 | Alive |
| 6 | 53 | M | Lung | Synovial sarcoma | grade3 | SS18-SSX1 | 1 | 4 | 1.2 mg/m2 | 1.2 mg/m2 | SD | 2% | | 19 | PD | 582 | 989 | Dead |

FNCLCC, Fédération Nationale des Centres de Lutte Contre le Cancer; PFS, progression-free survival; OS, overall survival; M, male; F, female; NA, not available; PR, partial response; SD, stable disease; PD, progressive disease.

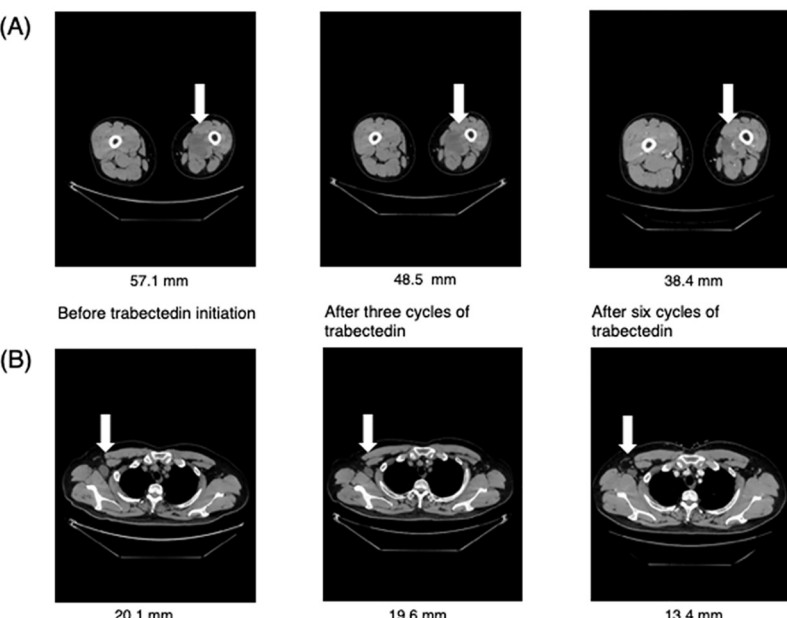

**Fig 2. Radiological response following three and six cycles of trabectedin in patient ID-4.** Lesions in the left thigh (**A**) and right axillary lymph node (**B**) are presented in the following order (left column, before trabectedin initiation; middle column, after three cycles of trabectedin; right column, after six cycles of trabectedin). The length of the left thigh lesion and the breadth of the right axillary lymph node are shown.

The clinical course of one patient with myxoid liposarcoma (patient ID-4) is as follows. The patient experienced relapse in the left lower extremity and underwent repeat surgical resection, radiotherapy, and a total of four lines of chemotherapy (doxorubicin, ifosfamide/etoposide, gemcitabine/docetaxel, and pazopanib). As a fifth regimen, trabectedin was initiated at a dose of 1.2 mg/m$^2$ in a tri-weekly cycle. Due to persistent myelosuppression, dose reduction and prolongation of the treatment interval were performed (cycle 1: 1.2 mg/m$^2$, 3-week interval; cycle 2: 1.0 mg/m$^2$, 3-week interval; cycle 3–6: 1.0 mg/m$^2$, 4-week interval; cycle 7: 0.8 mg/m$^2$, 4-week interval). No obvious hepatic impairment was observed. As shown in Fig 2, the radiological responses following three and six cycles were SD and PR, respectively. CT following 26 cycles showed maintained PR, and trabectedin administration was underway.

## Discussion

We reviewed 51 patients with sarcoma who had received trabectedin; 12% received over 1 year of treatment without disease progression. The final administration dose in two patients who continued the treatment over 20 cycles was the minimum dose (0.8 mg/m$^2$), and the treatment interval exceeded the duration mentioned in Food and Drug Administration and European Medicines Agency labels (3 weeks) [11, 12]. There existed a certain interval (median, approximately 6 months) from trabectedin initiation to radiological response in responders. These results suggest the validity of trabectedin continuation, even at lower doses or longer duration, even if tumor shrinkage is not observed immediately. Clinicians should consider both the efficacy and toxicity of the agent and administer it at a sustainable dose and interval for suitable cases. However, the treatment line or FNCLCC grade did not have an impact on PFS.

Furthermore, the GEISTRA score was not predictive, partly due to the low percentage of high-score cases or the limited sample size.

Several case reports have described the long-term response to trabectedin [13–16]. Davis et al. retrospectively compared clinical outcomes between 268 patients with sarcoma who had received trabectedin for 6–12 months and 133 patients who had received the treatment over 12 months [17]; the reported ORR and median OS in both treatment groups were 8% and 7%, and 18.1 months and 47.0 months, respectively. Two patients (one each with synovial sarcoma and uterine leiomyosarcoma) underwent the treatment for over 50 months. As with our data, patients who required cycle delay and dose reduction accounted for 62% and 78% of the >12 months group. Previous studies support the unique efficacy of trabectedin. According to a pooled analysis of two phase II trials, the ORR was 14% in the entire cohort and 27% in the myxoid liposarcoma cohort [18]. In their study, the duration from trabectedin initiation to the first response was approximately 4 months, and 35% of the patients received the agent for over 6 months.

Trabectedin exerts its activity on both tumor cells and the tumor microenvironment by reducing the production of inflammatory mediators (e.g., CCL2 or CXCL8) [19]. This may be a possible mechanism of slow tumor shrinkage. A preclinical study showed that deficiency in homologous recombination repair is an important factor for trabectedin sensitivity [20]. Tercero et al. immunohistochemically showed that high levels of XPG and low levels of BRCA2 and RAD51 expression are markers of better prognosis [21]. Schöffski et al. associated low levels of BRCA1 and high levels of ERCC1 and/or XPG using quantitative real-time polymerase chain reaction with improved prognosis [22]. In our study, three patients harbored no *BRCA1*, *BRCA2*, or *RAD51* mutations.

Another explanation for slow tumor regression may involve the pathological features of specific sarcoma subtypes. Regarding myxoid liposarcoma, which has myxoid stroma with varying degrees of cellularity, absorption of the myxoid stromal component may proceed gradually. These stromal changes may be observed as a late-onset tumor response by radiographic imaging modalities such as CT [18]. Interestingly, our patient with Ewing sarcoma maintained treatment for approximately 21 months. This fusion-driven malignancy typically exhibits aggressive clinical behavior and is generally considered unsuitable for trabectedin. Though clinical data involving trabectedin use for Ewing sarcoma is rare, a Children's Oncology Group phase II study with a Ewing sarcoma cohort demonstrated an ORR of 0% and a disease control rate of 10% [23]. Although EWSR1-FLI1 was originally considered a potential target for trabectedin, this has not been verified clinically, and further research is required [24].

This study had several limitations. First, it was retrospective, with a limited sample size. Second, the treatment choice after trabectedin initiation was chiefly dependent on each clinician's judgment, and the adopted regimen varied. Therefore, the direct impact of trabectedin could not be assessed, particularly on OS. Trabectedin dose or interval modification was also performed based on clinical judgment, and rigorous assessment of the association between treatment intensity and prognosis was difficult. Third, the image inspection interval varied in each patient, which might have affected the assessment of PFS. Fourth, the recommended dose is different between the US and Europe ($1.5 \text{ mg/m}^2$) and Japan ($1.2 \text{ mg/m}^2$). This may affect the clinical efficacy, although a phase 1 study in Japanese patients showed an area under the curve of $1.2 \text{ mg/m}^2$, similar to the $1.5 \text{ mg/m}^2$ reported in Caucasian patients. [11, 12, 25]. Furthermore, the FNCLCC grades in our study were based on pathology reports, which were unavailable for 14 (27%) patients. Finally, NGS data were available for only a few patients (6%), which complicated the finding of any gene mutations in relation to treatment response.

## Conclusions

Trabectedin continuation at a sustainable dose and interval is clinically beneficial in a subset of sarcoma patients, even at substantially lower doses and longer dosing intervals than the current standards. The time to tumor shrinkage is relatively long, particularly in myxoid liposarcoma patients. These results may be useful when considering the clinical application of trabectedin.

## Supporting information

**S1 Appendix.**
(XLSX)

## Acknowledgments

We wish to thank all the patients who participated in this study. We would also like to thank Editage (www.editage.com) for the English language editing.

## Author Contributions

**Conceptualization:** Akihiro Ohmoto, Kenji Nakano, Shunji Takahashi.

**Data curation:** Akihiro Ohmoto.

**Formal analysis:** Akihiro Ohmoto.

**Investigation:** Akihiro Ohmoto.

**Methodology:** Akihiro Ohmoto.

**Project administration:** Akihiro Ohmoto.

**Software:** Akihiro Ohmoto.

**Supervision:** Kenji Nakano, Shunji Takahashi.

**Validation:** Akihiro Ohmoto, Kenji Nakano, Naoki Fukuda, Xiaofei Wang, Tetsuya Urasaki, Naomi Hayashi, Hirotaka Suto, Shohei Udagawa, Ryosuke Oki, Yasuyoshi Sato, Mayu Yunokawa, Makiko Ono, Masanori Saito, Yusuke Minami, Keiko Hayakawa, Taisuke Tanizawa, Keisuke Ae, Seiichi Matsumoto, Junichi Tomomatsu, Shunji Takahashi.

**Visualization:** Akihiro Ohmoto.

**Writing – original draft:** Akihiro Ohmoto.

**Writing – review & editing:** Akihiro Ohmoto, Naoki Fukuda, Xiaofei Wang, Tetsuya Urasaki, Naomi Hayashi, Hirotaka Suto, Shohei Udagawa, Ryosuke Oki, Yasuyoshi Sato, Mayu Yunokawa, Makiko Ono, Masanori Saito, Yusuke Minami, Keiko Hayakawa, Taisuke Tanizawa, Keisuke Ae, Seiichi Matsumoto, Junichi Tomomatsu, Shunji Takahashi.

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
