## [Decision Letter · Decision Letter 0]

30 Jun 2022

PONE-D-22-07173Clinical characteristics of sarcoma cases in which long-term disease control was achieved with trabectedin treatment: A retrospective studyPLOS ONE

Dear Dr. Takahashi,

Thank you for submitting your manuscript to PLOS ONE. After careful consideration, we feel that it has merit but does not fully meet PLOS ONE’s publication criteria as it currently stands. Therefore, we invite you to submit a revised version of the manuscript that addresses the points raised during the review process.

We look forward to receiving your revised manuscript.

Kind regards,

Robert S Benjamin

Academic Editor

PLOS ONE

Journal Requirements:

2. Please provide additional details regarding participant consent. In the Methods section, please ensure that you have specified (1) whether consent was informed and (2) what type you obtained (for instance, written or verbal). If your study included minors, state whether you obtained consent from parents or guardians. If the need for consent was waived by the ethics committee, please include this information

Additional Editor Comments (if provided):

This manuscript describes a group of patients with prolonged disease control from trabectedin.

Reviewers' comments:

Reviewer's Responses to Questions

**Comments to the Author**

1. Is the manuscript technically sound, and do the data support the conclusions?

Reviewer #1: Yes

Reviewer #2: Yes

2. Has the statistical analysis been performed appropriately and rigorously? 

Reviewer #1: Yes

Reviewer #2: Yes

3. Have the authors made all data underlying the findings in their manuscript fully available?

Reviewer #1: Yes

Reviewer #2: Yes

4. Is the manuscript presented in an intelligible fashion and written in standard English?

Reviewer #1: Yes

Reviewer #2: Yes

5. Review Comments to the Author

Reviewer #1: This is a nicely written, albeit relatively small single-institution study evaluating the use of trabectedin in sarcoma. Overall, it provides useful, real-world information about the value of trabectedin long term in a subset of patients. This would be amplified if the authors pooled their data with one or more additional institutions. Since it was single-institution, there would theoretically be the opportunity to do additional tissue-based studies to look for some of the previously described tumor markers of interest. A few additional minor comments:

- Page 3, line 53- The authors mention that trabectedin is examined especially in translocation associated sarcoma, but do not explain the rationale. This should be clearly explained to the reader unfamiliar with the reasoning.

- Page 9- Since a main focus of this manuscript is the long-term patients, it would be helpful to include as much information as possible. Can the authors verify if patients had demonstrated progression prior to treatment? Amongst those with stable diseases, can a measurement of best tumor response/tumor shrinkage be included for these patients or a waterfall plot of all patients? As there are some patients with those histologies who can demonstrate an indolent course, in would be helpful to validate that there was benefit here.

- An additional reference that could be included was only available in abstract form but is relevant and could be added. Presented at CTOS: Davis EJ, et al. Abstract 277. Presented at: Connective Tissue Oncology Society Annual Meeting; Nov. 8-11, 2017; Maui.. Also presented at ESMO: Annals of Oncology (2017) 28 (suppl_5): v521-v538. 10.1093/annonc/mdx387

- The discussion section is very nicely written in pointing out the limitations of this analysis as well as the value of trabectedin in a subset. Congratulations to the authors.

- Table 2, typo in legend- overall survival

- Figure 1 is slightly blurry in reproduction.

Reviewer #2: Based on the Results section and Table 1, common histologies (myxoid liposarcoma, synovial sarcoma, leiomyosarcoma, and dedifferentiated liposarcoma) included 34 (67%) of the 51 patients. Would it be possible to report the rest as “other” or provide the other sarcomas as a foot note for Table 1?

If 51 patients were treated, why do we only have FNCLCC grades for 37 of 51 patients listed in the first paragraph of the Results section and on Table 1?

If 51 patients were treated and 26 had fusions, shouldn’t the percentage be 51% and not 53%? Is the 53% due to the number of evaluable patients (n=49), as stated in the Clinical outcomes section? It may be useful to clarify this point.

I assume that GEISTRA score was only assessed for 80% of patients based on the results section and Table 1 (Score 0 n=18, Score 1 n=23). Is this correct? Was there missing information for the remaining 20%?. It’s okay if some did not have sufficient information, but it would be good for this information to be clearly stated in the results section.

In the Clinical outcomes section, line 142 states “…among cases with GEISTRA scores 1, 2, and 3…” The results section/table 1 does not include any numbers for GEISTRA score 3.

6. PLOS authors have the option to publish the peer review history of their article (what does this mean?). If published, this will include your full peer review and any attached files.

Reviewer #1: No

Reviewer #2: **Yes: **Anthony Paul Conley

---

## [Author Response · Author response to Decision Letter 0]

17 Jul 2022

(Reviewer #1)

Comment 1. This is a nicely written, albeit relatively small single-institution study evaluating the use of trabectedin in sarcoma. Overall, it provides useful, real-world information about the value of trabectedin long term in a subset of patients. This would be amplified if the authors pooled their data with one or more additional institutions. Since it was single-institution, there would theoretically be the opportunity to do additional tissue-based studies to look for some of the previously described tumor markers of interest. A few additional minor comments: 

Response: We deeply appreciate the warm and suggestive comment.

Comment 2. Page 3, line 53- The authors mention that trabectedin is examined especially in translocation associated sarcoma, but do not explain the rationale. This should be clearly explained to the reader unfamiliar with the reasoning. 

Response: Thank you for the comment. As suggested, we added one sentence explaining the rationale as follows:

(Line 53-55) This agent has been examined especially for translocation-related sarcoma because trabectedin mechanistically suppresses the transcription of the oncogenic fusion proteins such as FUS-DDIT.

Comment 3. Page 9- Since a main focus of this manuscript is the long-term patients, it would be helpful to include as much information as possible. Can the authors verify if patients had demonstrated progression prior to treatment? Amongst those with stable diseases, can a measurement of best tumor response/tumor shrinkage be included for these patients or a waterfall plot of all patients? As there are some patients with those histologies who can demonstrate an indolent course, in would be helpful to validate that there was benefit here.

Response: Thank you for the essential comment. At first, we verified that all patients experienced disease progression prior to trabectedin using CT. Secondly, we summarized the maximum tumor response/tumor shrinkage rate from the baseline in Table 2. Thirdly, subtypes in patients who achieved long-term disease control (PFS > 1-year) are shown in Table 2. Except for small round cell sarcomas such as Ewing sarcoma, tumor differentiation score 1-2 in FNCLCC grade might reflect an indolent clinical behavior. In the context, we consider that 4 patients (patient ID 1,3,4,5) with tumor differentiation score 2 are cases with histologies who can demonstrate an indolent course. In terms of actual tumor volume, patient ID-6 had a single lung metastasis with a diameter of about 9 mm, which indicates an indolent behavior.

(Line 160-161) All 6 patients had confirmed disease progression prior to trabectedin initiation using computed tomography (CT).

Comment 4. An additional reference that could be included was only available in abstract form but is relevant and could be added. Presented at CTOS: Davis EJ, et al. Abstract 277. Presented at: Connective Tissue Oncology Society Annual Meeting; Nov. 8-11, 2017; Maui.. Also presented at ESMO: Annals of Oncology (2017) 28 (suppl_5): v521-v538. 10.1093/annonc/mdx387

Response: Thank you for the informative comment. We cited the reference at the part of discussion as follows:

(Line 217-224) Davis et al. retrospectively compared clinical outcomes between 268 patients with sarcoma who received trabectedin for 6-12 months and 133 patients who received the treatment over 12 months. Reported ORR and median OS in the both treatment groups were 8% and 7%, and 18.1 months and 47.0 months, respectively. Two patients (1 with synovial sarcoma and 1 with uterine leiomyosarcoma) underwent the treatment over 50 months. As with our data, patients who required cycle delay and dose reduction accounted for 62% and 78% of > 12 months group.

Comment 5. The discussion section is very nicely written in pointing out the limitations of this analysis as well as the value of trabectedin in a subset. Congratulations to the authors.

Response: Thank you so much for the kind comment.

Comment 6. Table 2, typo in legend- overall survival

Response: Thank you for the comment. We corrected the word in legend.

Comment 7. Figure 1 is slightly blurry in reproduction.

Response: Thank you for the comment. We corrected Figure 1.

(Reviewer #2)

Comment 1. Based on the Results section and Table 1, common histologies (myxoid liposarcoma, synovial sarcoma, leiomyosarcoma, and dedifferentiated liposarcoma) included 34 (67%) of the 51 patients. Would it be possible to report the rest as “other” or provide the other sarcomas as a foot note for Table 1?

Response: Thank you for the comment. As suggested, we added chondrosarcoma (n=4) and undifferentiated pleomorphic sarcoma (n=3) to the list in Table 1, and described the detailed histology information in a foot note as follows: 

*Other histology: alveolar soft part sarcoma (n=2), endometrial stromal sarcoma (n=2), rhabdomyosarcoma (n=1), Ewing sarcoma (n=1), clear cell sarcoma (n=1), spindle cell sarcoma (n=1), undifferentiated endometrial sarcoma (n=1), undifferentiated sarcoma (n=1) 

Comment 2. If 51 patients were treated, why do we only have FNCLCC grades for 37 of 51 patients listed in the first paragraph of the Results section and on Table 1?

Response: Thank you for the great comment. FNCLCC grades in our study were based on the pathology report in respective patient. Unfortunately, they were unavailable in 14 patients. Regarding to 3 out of the 14 patients, Japanese grade composed of tumor differentiation, tumor necrosis and Ki-67 index (not mitotic count) was alternatively obtained (Hum Pathol 2002; 33: 111-5). Two patients had grade 3 and 1 had grade 2. We clarified this point in the limitation, and gave an additional explanation in a foot note.

(Line 259-261) Furthermore, FNCLCC grades in our study were based on the pathology report in respective patient, and they were unavailable in 14 patients (27%).

**Japanese grade composed of tumor differentiation, tumor necrosis and MIB-1 index was available in other 3 patients. Grade 2 (n=1), grade 3 (n=2)

Comment 3. If 51 patients were treated and 26 had fusions, shouldn’t the percentage be 51% and not 53%? Is the 53% due to the number of evaluable patients (n=49), as stated in the Clinical outcomes section? It may be useful to clarify this point.

Response: Thank you for the comment. We corrected the percentage to 51% (26/51) in the main text and Table 1.

Comment 4. I assume that GEISTRA score was only assessed for 80% of patients based on the results section and Table 1 (Score 0 n=18, Score 1 n=23). Is this correct? Was there missing information for the remaining 20%?. It’s okay if some did not have sufficient information, but it would be good for this information to be clearly stated in the results section.

Response: Thank you for the comment. I am sorry for the confusion. We checked the original data again. As written in line 111-112, GEISTRA scores were 0 (n=18), 1 (n=23), and 2 (n=10). There were no cases with GEISTRA score 3. We added the missing information in Table 1.

Comment 5. In the Clinical outcomes section, line 142 states “…among cases with GEISTRA scores 1, 2, and 3…” The results section/table 1 does not include any numbers for GEISTRA score 3.

Response: Thank you for the comment. As we responded to comment 4, there were no cases with GEISTRA score 3. Therefore, we corrected the expression in line 142 as follows:

“among cases with GEISTRA scores 0, 1, and 2 (P= 0.61)”

(Journal Requirements)

Response: We modified our manuscript to PLOS ONE's style.

2. Please provide additional details regarding participant consent. In the Methods section, please ensure that you have specified (1) whether consent was informed and (2) what type you obtained (for instance, written or verbal). If your study included minors, state whether you obtained consent from parents or guardians. If the need for consent was waived by the ethics committee, please include this information

Response: We described additional details regarding participant consent.

(Line 75-76)

Informed consent of each patient was waived under the approval of the institutional review board.

3. In your Data Availability statement, you have not specified where the minimal data set underlying the results described in your manuscript can be found. PLOS defines a study's minimal data set as the underlying data used to reach the conclusions drawn in the manuscript and any additional data required to replicate the reported study findings in their entirety.

Response: We added my data availability statement in line 278-279.

---

## [Decision Letter · Decision Letter 1]

26 Sep 2022

PONE-D-22-07173R1Clinical characteristics of sarcoma cases in which long-term disease control was achieved with trabectedin treatment: A retrospective studyPLOS ONE

Dear Dr. Takahashi,

Thank you for submitting your manuscript to PLOS ONE. After careful consideration, we feel that it has merit but does not fully meet PLOS ONE’s publication criteria as it currently stands. Therefore, we invite you to submit a revised version of the manuscript that addresses the points raised during the review process. The scientific issues raised on initial review of the manuscript have been addressed to the satisfaction of both reviewers.  One reviewer noted a number of stylistic improvements that could be made after thorough reviewof your submission by a native English speaker.  Unfortunately, PLOS ONE does not provide such a service and we encourage you to find additional assistance locally. Please submit your revised manuscript by Nov 10 2022 11:59PM. If you will need more time than this to complete your revisions, please reply to this message or contact the journal office at plosone@plos.org. Please include the following items when submitting your revised manuscript:A rebuttal letter that responds to each point raised by the academic editor and reviewer(s). You should upload this letter as a separate file labeled 'Response to Reviewers'.A marked-up copy of your manuscript that highlights changes made to the original version. You should upload this as a separate file labeled 'Revised Manuscript with Track Changes'.An unmarked version of your revised paper without tracked changes. You should upload this as a separate file labeled 'Manuscript'.If applicable, we recommend that you deposit your laboratory protocols in protocols.io to enhance the reproducibility of your results. Protocols.io assigns your protocol its own identifier (DOI) so that it can be cited independently in the future. For instructions see: https://journals.plos.org/plosone/s/submission-guidelines#loc-laboratory-protocols. Additionally, PLOS ONE offers an option for publishing peer-reviewed Lab Protocol articles, which describe protocols hosted on protocols.io. Read more information on sharing protocols at https://plos.org/protocols?utm_medium=editorial-email&utm_source=authorletters&utm_campaign=protocols.

We look forward to receiving your revised manuscript.

Kind regards,

Robert S Benjamin

Academic Editor

PLOS ONE

Journal Requirements:

Reviewers' comments:

Reviewer's Responses to Questions

**Comments to the Author**

1. If the authors have adequately addressed your comments raised in a previous round of review and you feel that this manuscript is now acceptable for publication, you may indicate that here to bypass the “Comments to the Author” section, enter your conflict of interest statement in the “Confidential to Editor” section, and submit your "Accept" recommendation.

Reviewer #1: (No Response)

2. Is the manuscript technically sound, and do the data support the conclusions?

Reviewer #1: Yes

3. Has the statistical analysis been performed appropriately and rigorously? 

Reviewer #1: Yes

4. Have the authors made all data underlying the findings in their manuscript fully available?

Reviewer #1: Yes

5. Is the manuscript presented in an intelligible fashion and written in standard English?

Reviewer #1: Yes

6. Review Comments to the Author

Reviewer #1: The authors addressed comments and once again put forward an informative manuscript.

My only remaining comment is more general. While the article relays good information, certain stylistic concerns remain in a scientific paper. Please consider having article briefly edited for scientific English language to improve on conciseness to better communicate the authors' points. It is noted that a program was used for this purpose, but a native speaker may provide more assistance. A few examples are provided:

Page 3, line 66; "It is very important to predict who can continue treatment sufficiently before trabectedin is administered; yet, there is insufficient clinical information regarding this point." could easily be adjusted to "Despite the importance, there are few known clinical predictive factors of benefit to trabectedin" or something like that.

Page 13, line 247, "Originally, a specific translocation (EWSR1-FLI1) could be the target of trabectedin, and further investigations are awaited [24]". This line could be adjusted minimally to enhance understanding to "While EWSR1-FLI1 was originally hypothesized to be a potential target for trabectedin, this has not been seen clinically and more studies are needed" or something similar.

Finally, would suggest minor adjustments to Conclusions. Currently reads as " This study suggests that trabectedin continuation at a sustainable dose and interval might be clinically beneficial in some patients with a focus on L-sarcomas. Specifically, a longer duration is expected for myxoid liposarcoma since the tumor shrinkage effect is generally slow." This may better relay the author's main points as "The study suggests that trabectedin continuation at a sustainable dose and interval is clinically beneficial in a subset of sarcoma patients, even if substantially lower dosing and longer dosing intervals than current standards. The time to tumor shrinkage is relatively long, particularly in patients with myxoid liposarcoma."

There are several other instances where a quick read through to adjust language can help with readability. Beyond that, the authors have done an excellent job trying to relay their work..

7. PLOS authors have the option to publish the peer review history of their article (what does this mean?). If published, this will include your full peer review and any attached files.

Reviewer #1: No

---

## [Author Response · Author response to Decision Letter 1]

29 Oct 2022

(Reviewer #1)

Comment 1. The authors addressed comments and once again put forward an informative manuscript. My only remaining comment is more general. While the article relays good information, certain stylistic concerns remain in a scientific paper. Please consider having article briefly edited for scientific English language to improve on conciseness to better communicate the authors' points. It is noted that a program was used for this purpose, but a native speaker may provide more assistance. 

Response: Thank you for the informative comment. We consulted with the professional English proofreading company again, and a native English editor modified throughout my original manuscript. 

Comment 2. Page 3, line 66; "It is very important to predict who can continue treatment sufficiently before trabectedin is administered; yet, there is insufficient clinical information regarding this point." could easily be adjusted to "Despite the importance, there are few known clinical predictive factors of benefit to trabectedin" or something like that.

Response: Thank you for the comment. As suggested, we corrected the original sentence (Page 3, line 64).

Despite its importance, few clinical predictive factors of trabectedin’s benefits are known.

Comment 3. Page 13, line 247, "Originally, a specific translocation (EWSR1-FLI1) could be the target of trabectedin, and further investigations are awaited [24]". This line could be adjusted minimally to enhance understanding to "While EWSR1-FLI1 was originally hypothesized to be a potential target for trabectedin, this has not been seen clinically and more studies are needed" or something similar.

Response: Thank you for the comment. As suggested, we corrected the original sentence (Page 13, line 240-241).

Although EWSR1-FLI1 was originally considered a potential target for trabectedin, this has not been verified clinically, and further research is required.

Comment 4. Finally, would suggest minor adjustments to Conclusions. Currently reads as " This study suggests that trabectedin continuation at a sustainable dose and interval might be clinically beneficial in some patients with a focus on L-sarcomas. Specifically, a longer duration is expected for myxoid liposarcoma since the tumor shrinkage effect is generally slow." This may better relay the author's main points as "The study suggests that trabectedin continuation at a sustainable dose and interval is clinically beneficial in a subset of sarcoma patients, even if substantially lower dosing and longer dosing intervals than current standards. The time to tumor shrinkage is relatively long, particularly in patients with myxoid liposarcoma."

Response: Thank you for the great comment. As suggested, we corrected the original sentence (Page 14, line 258-261).

Trabectedin continuation at a sustainable dose and interval is clinically beneficial in a subset of sarcoma patients, even at substantially lower doses and longer dosing intervals than the current standards. The time to tumor shrinkage is relatively long, particularly in myxoid liposarcoma patients.

Comment 5. There are several other instances where a quick read through to adjust language can help with readability. Beyond that, the authors have done an excellent job trying to relay their work..

Thank you for the comment. A native English editor modified throughout my original manuscript. These points are highlighted in yellow. We attach Certificate of Proofreading as a supplementary document.

---

## [Decision Letter · Decision Letter 2]

3 Jan 2023

Clinical characteristics of sarcoma cases in which long-term disease control was achieved with trabectedin treatment: A retrospective study

PONE-D-22-07173R2

Dear Dr. Takahashi,

We’re pleased to inform you that your manuscript has been judged scientifically suitable for publication and will be formally accepted for publication once it meets all outstanding technical requirements.

Kind regards,

Robert S Benjamin

Academic Editor

PLOS ONE

Additional Editor Comments (optional):

While the current version of the manuscript is acceptable for publication, it would be further improved by minor revisions suggested by reviewer 2. I would suggest incorporation of those changes into the final manuscript, but further external review is not necessary.

Reviewers' comments:

Reviewer's Responses to Questions

**Comments to the Author**

1. If the authors have adequately addressed your comments raised in a previous round of review and you feel that this manuscript is now acceptable for publication, you may indicate that here to bypass the “Comments to the Author” section, enter your conflict of interest statement in the “Confidential to Editor” section, and submit your "Accept" recommendation.

Reviewer #1: All comments have been addressed

Reviewer #2: All comments have been addressed

2. Is the manuscript technically sound, and do the data support the conclusions?

Reviewer #1: Yes

Reviewer #2: Yes

3. Has the statistical analysis been performed appropriately and rigorously? 

Reviewer #1: Yes

Reviewer #2: Yes

4. Have the authors made all data underlying the findings in their manuscript fully available?

Reviewer #1: Yes

Reviewer #2: Yes

5. Is the manuscript presented in an intelligible fashion and written in standard English?

Reviewer #1: Yes

Reviewer #2: Yes

6. Review Comments to the Author

Reviewer #1: The authors did an excellent job revising paper such that it reads well and relays main points clearly.

Reviewer #2: I have no issues with the content of the manuscript. I do believe this paper will provide valuable insight regarding the use of trabectedin for sarcoma. I would like to offer some suggested grammatical edits:

Page 3, line 65: We previously published our clinical experience of 38 patients with advanced sarcoma treated with trabectedin (ref 7). Our current study includes a larger sample size, and it highlights the clinical features of patients with long-term disease control achieved with trabectedin.

Page 4, line 71: This study was approved by the ...

Page 4, line 75-76: ...extracted the data of 51 advanced sarcoma patients who received trabectedin.

Page 4, line 83: ...our analysis included patients who received it as ....

Page 4, line 84: ...and as part of a clinical trial prior to regulatory approval.

Page 5, line 106-107: Common histological subtypes included myxoid liposarcoma...

Page 11, line 197-198: ...received trabectedin; 12% received over 1 year of treatment without disease progression.

Page 12, line 221-222: Line 222 could be the start of a new paragraph.

Page 12, line 223: This may be a possible mechanism of...

Page 13 line 231-232: Another explanation for slow tumor regression may involve the pathological features of specific sarcoma subtypes.

Page 13, line 233-235: ...absorption of the myxoid stromal component may proceed gradually. These stromal changes may be observed as a late-onset tumor response by radiographic imaging modalities such as computed tomography.

Page 13, line 236: This fusion-driven malignancy typically exhibits aggressive clinical behavior and is generally considered unsuitable for trabectedin.

Page 13, line 237: Though clinical data involving trabectedin use for Ewing sarcoma is rare, a Children's Oncology Group phase II study with a Ewing sarcoma cohort demonstrated an ORR of 0% and a disease control rate of 10% (ref 23)

7. PLOS authors have the option to publish the peer review history of their article (what does this mean?). If published, this will include your full peer review and any attached files.

Reviewer #1: No

Reviewer #2: **Yes: **Anthony P Conley, MD

---

## [Editor Report · Acceptance letter]

5 Jan 2023

PONE-D-22-07173R2 

Clinical characteristics of sarcoma cases in which long-term disease control was achieved with trabectedin treatment: A retrospective study 

Dear Dr. Takahashi:

I'm pleased to inform you that your manuscript has been deemed suitable for publication in PLOS ONE. Congratulations! Your manuscript is now with our production department. 

Kind regards, 

on behalf of

Professor Robert S Benjamin 

Academic Editor

PLOS ONE